# Glycan Analysis as Biomarkers for Testicular Cancer

**DOI:** 10.3390/diagnostics9040156

**Published:** 2019-10-22

**Authors:** Michal Hires, Eduard Jane, Michal Mego, Michal Chovanec, Peter Kasak, Jan Tkac

**Affiliations:** 1Institute of Chemistry, Slovak Academy of Sciences, Dubravska cesta 9, 845 38 Bratislava, Slovakia; michal.hires@savba.sk (M.H.); eduard.jane58@gmail.com (E.J.); 2Translational Research Unit, Faculty of Medicine, Comenius University and National Cancer Institute, Klenova 1, 833 10 Bratislava, Slovakia; misomego@gmail.com (M.M.); michal.chovanec1@gmail.com (M.C.); 32nd Department of Oncology, Faculty of Medicine, Comenius University and National Cancer Institute, Klenova 1, 833 10 Bratislava, Slovakia; 4Center for Advanced Materials, Qatar University, Doha 2713, Qatar

**Keywords:** testicular cancer, glycosylation, lectins, glycans, biomarkers

## Abstract

The U.S. Preventive Services Task Force does not recommend routine screening for testicular cancer (TC) in asymptomatic men, essentially because serological testicular cancer (TC) biomarkers are not reliable. The main reason is that two of the most important TC biomarkers, α-fetoprotein (AFP) and human chorionic gonadotropin (hCG), are not produced solely due to TC. Moreover, up to 40% of patients with TC do not have elevated serological biomarkers, which is why serial imaging with CT is the chief means of monitoring progress. On the other hand, exposure to radiation can lead to an increased risk of secondary malignancies. This review provides the first comprehensive account of the applicability of protein glycoprofiling as a promising biomarker for TC with applications in disease diagnostics, monitoring and recurrence evaluation. The review first deals with the description and classification of TC. Secondly, the limitations of current TC biomarkers such as hCG, AFP and lactate dehydrogenase are provided together with an extensive overview of the glycosylation of hCG and AFP related to TC. The final part of the review summarises the potential of glycan changes on either hCG and AFP as TC biomarkers for diagnostics and prognostics purposes, and for disease recurrence evaluation. Finally, an analysis of glycans in serum and tissues as TC biomarkers is also provided.

## 1. Description, Risk Factors and Classification of Testicular Cancer (TC)

Cancer is either the first or second most common cause of death below the age of 70 in more than half of 172 countries, according to the estimates from the World Health Organisation in 2015 [1]. It is predicted that 9.6 million people died as a result of cancer and 18.1 million new cases were recorded worldwide in 2018 [1]. Testicular cancer (TC) is relatively rare, but it is the most common solid tumour found in young men [2,3].

### 1.1. TC Risk Factors

Recent studies suggest that TC is a result of interactions of multiple factors including environmental and genetic ones, a claim supported by the study involving monozygotic/dizygotic twins [4,5]. Disrupted foetal hormone signalling during foetal development due to the influence of various xenobiotics as well as misbalanced maternal hormone levels is thought to be the cause of multiple disorders linked to TC [6,7,8,9]. A congenital defect, when one or both testicles are undescended into the scrotum (cryptorchidism), is connected with 5-fold increase of TC risk [3,10,11]. Furthermore, inflammation of the testicles, for example, due to overcoming mumps and various testicular injuries also has a negative effect [6]. Positive family history might be a strong risk factor, since the risk of TC development increases when TC is diagnosed for direct blood relatives for men, like father and brother [12,13]. When TC is diagnosed in one testis, there is a risk of the disease developing in the opposite testis. The environmental impact on TC development and progression is supported by the study concluding that sons of men who moved from a low- to a high-incidence place have the same risk of TC as males living at a high-incidence place and the effect of harmful environmental factors can be observed already in the 2nd generation [14]. Some information indicates that postnatal, environmental and lifestyle (a diet and an exposure to endocrine disrupting agents) factors may negatively influence the development of TC [15].

### 1.2. TC Classification

The rapid and efficient diagnosis, as well as the prognosis, is highly complicated due to the high diversity of the disease. The most common first symptoms are changes in the area of the testicles. There can be gradually growing lumps of solid consistency with a smooth, sometimes bumpy surface that are detected by palpation randomly or during examination. Only 20% of patients suffer from pain, which is described as a blunt pain and only accidentally as a severe pain with a risk of a tumour bleeding. However, much more has been discovered by physicians in the investigation of patients who suffer from testis and prostate atrophy, feminization, gynecomastia, hair loss or libido change due to hormonal changes. TC is classified according to the histological tissue composition, the germ cell lineage and the age at the onset of TC [3,16]. More than 90–95% of TC are germ cell tumours (GCTs) affecting testicular germ cells (cells making sperms) [6,17]. The current WHO classification system defines two major entities of GCTs as germ cell neoplasia in situ (GCNIS)-related and non-GCNIS-related (non-GCNIS) (Figure 1) [3,16]. Approximately 60% of GCTs contain more than one type of histological pattern (i.e., mixed GCTs) [18].

The remaining TCs are of non-germ cell origin and include sex cord and gonadal stromal tumours, lymphoid and hematopoietic tumours and metastatic tumours from other primary neoplasms [17,19,20]. From pure non-germ cell tumours, the largest occurrence has Leydig cell tumour (LCT), then Sertoli cell tumour (SCT), granulosa cell tumour and pure stromal tumour [19,21].

## 2. TC Biomarkers and Their Limitations

The U.S. Preventive Services Task Force does not recommend routine screening for TC in asymptomatic men [1]. Scrotal ultrasonography is the initial diagnostic tool in TC screening. After a positive outcome from ultrasonography, when a solid intratesticular mass is discovered, orchiectomy is applied for both diagnostic and therapeutic purposes. Finally, a process of staging through chest radiography, chemistry panel, liver function tests and tumour markers guides the treatment of TC patients [22].

Since TC serological biomarkers are not reliable, it is essential to find a way to reliably diagnose TC as early as possible to monitor the course of treatment or to predict the disease development [5]. Two of the most important TC biomarkers, α-fetoprotein (AFP) and human chorionic gonadotropin (hCG), are not produced solely due to TC. The AFP level increases physiologically during the first two years of life, but also in gastrointestinal or liver tissue tumours [23,24].The hCG value may be elevated due to other cancer types, marijuana use, etc. [25,26,27]. In addition, these two biomarkers are limited to some types of TC (AFP for yolk sac tumour and hCG for TC choriocarcinoma) (Table 1). AFP and/or hCG are elevated in up to 70% of patients with non-seminomatous GCTs [28,29]. The third biomarker, albeit less informative, is lactate dehydrogenase (LDH). Further, hCG is elevated in 15–30% of seminoma patients at the time of diagnosis and LDH is elevated in up to 80% of patients with advanced metastatic seminoma [28,29].

The levels of LDH, AFP and hCG are factors for the risk stratification of TC based on the International Germ-Cell Cancer Collaborative Group classification [30]. It is known that up to 40% of patients with TC do not exhibit elevated AFP, hCG or LDH levels. For these patients, serial imaging with CT is the main means of monitoring the progress of the disease. On the other hand, a radiation load can lead to an increased risk of secondary malignancies [31]. Accordingly, it is important to find biomarkers that meet the following criteria: (1) adequate half-life with stability within the bloodstream; (2) the presence in blood/urine in sufficient concentration; and (3) the presence in all TC types, irrespective of age, the localisation and histological profile [32]. A full list of what is required for the biomolecules to become a biomarker can be found elsewhere [33].

### 2.1. Human Chorionic Gonadotropin (hCG)

The protein, hCG, was discovered to be a biomarker of TC in 1938 [34]. Both subunits of hCG (37 kDa) are heavily glycosylated (Table 1) with one third of the molecular mass of hCG consisting of glycans [35]. Subunit α (92 amino acids) contains two *N*-glycans (Asn52 and Asn78) (Figure 2a) and exhibits similarity with the follicle-stimulating hormone, luteinising hormone and thyroid-stimulating hormone [18,36,37,38,39]. The larger subunit β (145 amino acids) contains two *N*-glycans (Asn13 and Asn30) and 4 *O*-glycans (Ser121, Ser127, Ser132 and Ser138) (Figure 2a). Of those 4 *O*-glycans, three are type 1 *O*-glycans (Ser127, Ser132, and Ser138) and one is a type 2 *O*-glycan (Ser121) (Figure 2a). hCG and some of its forms are one of the most acidic proteins in a human body with pI of 3.5 for hCG and pI of 3.2 for a hyperglycosylated hCG [40,41].

Hyperglycosylated hCG (hCG-H) is here defined as an hCG form recognised by the antibody B152, i.e., having type 2 *O*-glycan on Ser132 of the β-subunit of hCG [42]. The malignancy-associated hCG-H or hCGβ-H (β-subunit of hCG-H) carrying triantennary *N*-glycan at Asn30, fucosylation at Asn13 and core 2 *O*-glycans at Ser127, Ser132 and/or Ser138 (Figure 2b) [42] was discovered by Professor Cole in 1997 [43].

Hyperglycosylated hCG forms are not only indicators of the presence of cancer, but also enhance both cancer growth and cell invasion when supplemented into cell lines [44,45]. Since hCGβ-H and hCG-H induce the malignancy of different cancer types, the B152 antibody recognising these two forms of hCG appears to be a promising biomolecule applicable to the therapeutic treatment of the disease [41]. Hyperglycosylated forms of hCG are currently defined as drivers for most or possibly all human cancers [46]. 

The protein, hCG and its various forms can be divided into several groups:(A)two types when considering the binding preference [41]:binding to LH/hCG hormone receptor (hormone hCG, a normal form of hCG);binding to TGFβ-II receptor (hCG-H or hCGβ-H).(B)four molecules, when considering the glycosylation pattern containing [47,48]:four biantennary *N*-glycans, three type 1 *O*-glycans (tri- and tetra-saccharides), one type 2 O-glycan (hCG);four biantennary *N*-glycans and four type 2 *O*-glycans (penta- and hexa-saccharides) (placental hyperglycosylated hCG);triantennary *N*-glycans on β-subunit and type 2 *O*-glycans (extravillous cytotrophoblast hyperglycosylated hCG and cancer hyperglycosylated hCG);*N*-acetylgalactosamine-sulphate terminating *N*-glycans and type 1 *O*-glycans (sulphated hCG hormone).(C)seven semi-independent molecules, when considering functional roles [41]: placental hCG (controlling pregnancy);placental autocrine hyperglycosylated hCG (start placentation during pregnancy);pituitary sulphated hCG (present during ovarian steroidogenesis, ovulation & luteogenesis);foetal hCG hormone (promoting foetal organ growth);ovarian hyperglycosylated hCG (driving the final proteolytic enzymatic step during ovulation);hCG-H (trophoblastic tumours);hCGβ-H (non-trophoblastic tumours).(D)significant structural variability involving numerous isoforms combining glycosylation (14 major variants for hCGα and 12 major variants for hCGβ) [35,49,50] and structural (nicked forms of hCG, hCGβ, hCG-H, hCGβ-H; hCG forms without a terminal CTP tail (β113-145 containing all 4 *O*-glycans); β-core fragment (β6-40 and β55-92 linked by a disulphide bond)) variations [35,41].

Hyperglycosylation (a change of type 1 *O*-glycans into type 2 *O*-glycans) affects protein-folding, resulting in the exposure of an otherwise hidden sequence, which is nicked/cleaved at β47-48 by leukocyte elastase resulting in the dissociation of hCG-H into hCGβ-H [41]. While hyperglycosylated hCG cannot bind to TGFβ-II receptors directly, hyperglycosylated hCGβ is able to do so [41]. It has been shown that molecules acting on TGFβ-II receptors induce the production of metalloproteinases and collagenases and thereby promote cell-to-cell invasion [51]. Multiple studies have also shown that all malignancies produce either hCGβ (non-trophoblastic cancers) or hyperglycosylated hCG (trophoblastic cancers including testicular GCTs) [51,52,53]. Non-trophoblastic cancers do not produce a dimeric form of hCG, due to the absence of an enzyme, disulphide isomerase, which adds two disulphide bonds to the β subunit and only this subunit can be recognised by the α subunit to form a dimer [41]. The protein is present in blood and urine in at least 18 forms and degradation products of different sizes and levels of glycosylation [40]. The most recent study suggests the presence of a 4th hCG form, i.e., a sulphated hCG—hormone hCG with *N*-acetygalactosamine-sulphate terminating glycan [48]. In addition, there are indications that the automated tests currently in use in clinical laboratories to determine hCG levels are not optimal. They cannot distinguish between the different protein forms, or they cannot capture all of them [41].

### 2.2. α-Fetoprotein (AFP)

Alpha-fetoprotein (AFP; 70 kDa) is a glycoprotein composed of 591 amino acids with different characteristics, as shown in Table 1. The protein is normally synthesised in the yolk sac, liver and intestine and serves as a major serum-binding protein. AFP has 3 isoforms: L1 is produced in non-neoplastic liver disease, L2 is produced in yolk sac tumours, and L3 is produced in hepatocellular carcinoma and hepatoblastoma [54]. High levels are observed physiologically in the first years of life, in infective-degenerative liver diseases and during regeneration of the liver after toxic damage, in yolk sac tumours, embryonal carcinoma, but also in hepatocellular carcinoma (HCC). High biomarker levels are noted in 50–70% of patients with non-seminomas, thus AFP is the most commonly elevated tumour marker in TC [55]. Pure tumours (TC choriocarcinoma and seminoma) do not have the potential to produce AFP. The half-life of AFP is 5–7 days, which is approximately 5-times higher in comparison with hCG.

The serological AFP level is routinely used in diagnosis, therapy monitoring and in a follow-up process for patients with GCTs. The level of AFP during patient-monitoring, however, can be difficult to interpret since elevated AFP can result from non-tumour liver activity, such as hepatotoxicity due to chemotherapy [56]. 

### 2.3. Lactate Dehydrogenase (LDH)

Lactate dehydrogenase (LDH; 134 kDa) is a cytoplasmic enzyme produced by many types of tissues, including muscle (skeletal, smooth, cardiac), liver, kidney and brain [18,57,58]. LDH is a tetrameric protein composed of two structurally different subunits. In serum, the enzyme is present in a form of 5 isoenzymes, which concentration correlates with the number of gene copies in a short arm of chromosome 12p, where they are coded [18,37,59]. Isoenzyme LDH-1 is the most frequent form of LDH present during increased levels of LDH [18,37,60]. Although 40–60% of men with TC of germ cell etymology have elevated LDH, this marker may be considered as an alternative serological marker due to the relatively low specificity for GCTs compared to AFP and hCG [18,37,58,61]. hCG and AFP are produced by tumour cells, but an increased level of LDH in serum is a result of cell damage, as well [57]. Nonetheless, there are some correlations between LDH levels and the survival of TC patients [37,62]. LDH is one of the three sole serum markers currently used for risk stratification in TC, based on the International Germ Cell Consensus Classification (IGCCC) criteria [32,63]. LDH has a limited sensitivity and specificity for seminoma. It is increased in approximately 80% of advanced seminomas. An elevated LDH level was observed in 60% of advanced non-seminomas with a value higher than 2000 U/L, indicating an advanced disease and/or disease recurrence [18,58,64]. LDH assays determine activity not quantity, therefore the differences in values can be expected between several methods applied for its activity assays [18].

### 2.4. Other TC Biomarkers

A recent review made the following conclusion: “Having systematically reviewed the available literature, we found surprisingly little evidence to guide optimal testing with biomarkers (AFP, hCG and LDH) routinely used during follow-up for testicular cancer recurrence” [65]. Due to all these reasons, there is still a substantial effort to find novel biomarkers which are more reliable for example, for disease prognosis [66] or diagnostics (DNA methylation, microRNAs, proteins) [17,67,68,69,70]. 

Neuron-specific enolase, an isoenzyme of the glycolytic enzyme 2-phospho-d-glycerate-hydrolase, is elevated in approximately 30–50% of patients with seminoma, specifically in metastatic stages. Moreover, the protein level can be increased in patients with normal hCG and LDH concentration [17,71,72]. On the other hand, it is not a reliable marker due to the high false-positive rate [73]. Elevation was observed also in other conditions [24,74].

Another TC biomarker is a placental alkaline phosphatase. There are 2 genes coding the proteins with alkaline phosphatase activity, the placental (PLAP) and germ cell enzyme (also noted as a placental-like alkaline phosphatase) [75]. The enzyme is physiologically expressed in foetal germ cells and in infants [18]. Therefore, the staining results in the first years of postnatal life must be interpreted with caution [25]. The protein is also produced ectopically by a variety of malignant tumours [76,77]. The elevated protein level is observed in approximately 80% of TC patients [78] and the most frequently in seminoma TC (60–70%) [79]. Despite the low false-positive rates (1.6%), its potential for disease monitoring is complicated by the fact that its serum level can increase up to 10-fold by smoking [26,65,78,79].

TC cells express several high molecular weight glycoproteins. One of these antigens, sialylated keratin sulphate proteoglycan, can be detected by monoclonal antibody against TRA-1-60 (podocalyxin) [80,81]. It is expressed by embryonal cancer, seminoma and carcinoma in situ of the testis [82,83]. The study showed that the antigen is expressed in approximately 80% of patients with advanced embryonal carcinoma. Although its level decreases during chemotherapy, 15–30% patients do not have normalized levels after therapy [84]. A low assay specificity limits its wider use [17,18,37].

From novel TC biomarkers discovered thus far, the following biomolecules/cells can be listed: microRNAs [26,85], DNA methylation [86], circulating tumour cells [26], circulating DNA [26], various proteins [26]. DNA-based biomarkers can be used for non-invasive diagnostics due to the presence of different DNA types in blood stream [87]. An increased level of circulating tumour DNA was observed in men with TC and circulating tumour DNA can distinguish patients with cancer from healthy ones (88% sensitivity and 97% specificity), also in cases with a normal level of conventional markers [88,89]. DNA present in blood is produced by different organs/cells and this is why by DNA analysis, a clearer and more complex picture about the substantial heterogeneity of TC can be obtained. On the other hand, DNA is unstable in the blood stream and it is rapidly cleared. Therefore, the use of this marker for diagnosis requires high-throughput and sensitive techniques [17]. The largest potential from novel TC biomarkers have microRNAs associated with different types of TC [68]. MicroRNAs are highly stable versions of the RNA, modulating protein-coding genes expression. MicroRNAs act either as oncogenes or tumour-suppressor genes. In cancer, they are dysregulated and their profiles can show the origin of tumours. Due to these properties, microRNAs are promising biomarkers for cancer monitoring [32]. For example, TC patients negative for microRNA-371a-3p had a better progression-free survival and an overall survival compared to the TC patients with microRNA-371a-3p present in serum [90].

None of the mentioned markers are universal and specific. Recently, it was discovered that extracellular vesicles, such as exosomes, can be a rich source of various types of biomarkers as detected for various types of urological tumours [91]. As exosomes so far have not been applied for TC diagnostics and/or monitoring, their application in the discovery for novel and robust TC biomarkers is extremely exciting.

## 3. Glycans as TC Biomarkers

The changes/alterations in glycosylation can be successfully applied to the discovery of novel cancer-related biomarkers. It is estimated that 70+% of all proteins are post-translationally modified by glycosylation with the involvement of glycans in cancer development and progression [92,93,94,95,96,97,98]. A recent paper in Science showed that, for reliable and accurate diagnostics of cancer at an early stage, multiple analytes need to be determined in serum, including the levels of several proteins and cell-free DNA [99]. Although, in this pioneering study [99], glycan analysis was not implemented for cancer diagnostics, a forthcoming study showed that glycan analysis can deliver more reliable results with excellent discrimination between indolent localised prostate cancer and an aggressive non-localised form of the disease [100]. Accordingly, the sections below focus on an evaluation of the clinical performance characteristics of glycans as TC biomarkers.

### 3.1. Performance of Glycosylated hCG as a TC Biomarker

The performance of hyperglycosylated hCG was evaluated in order to discriminate between various types of cancer based on the data presented in Cole’s paper [51]. This is shown in the Appendix A (Appendix A). 

hCG from urine samples of pregnant women and patients having choriocarcinoma (*n* = 3), invasive mole (*n* = 3), male GCT (*n* = 2) and a non-pregnant control were glycoprofiled using several lectins [101]. The results showed that the following lectins were able to distinguish hCG from GCT patients when compared to the non-pregnant control: *Galanthus nivalis* agglutinin, *Phaseolus vulgaris* leukoagglutinin (recognising branched glycans), *Pisum sativum* agglutinin (recognising glucose/mannose), *Ricinus communis* agglutinin (recognising terminal galactose), *Maackia amurensis* agglutinin II (recognising 2,3-linked sialic acid), *Sambucus nigra* agglutinin (binding to 2,6-linked sialic acid) and wheat germ agglutinin (recognising sialic acid and β1,4-GlcNAc). The authors acknowledge that the glycan composition on hCG from cell lines, urine and serum might be different due to the partial hCG degradation during renal secretion [101]. This lectin-based glycoprofiling of hCG can complement the use of antibody B152, raised against type 2 *O*-glycan on Ser132 of hCG (hCG-H or hCGβ-H) [101].

hCGβ isolated from the urine of patients with non-seminomatous TC (*n* = 2, stages 1 and 3), choriocarcinoma (*n* = 1), invasive mole (*n* = 1), pregnant women (*n* = 2) and a choriocarcinoma cell line (*n* = 1) was applied to the analysis of site-specific glycan structures using liquid chromatography combined with mass spectrometry [102]. With regard to *N*-glycan analysis, triantennary glycans and fucosylation increased at Asn13 and Asn30 of hCGβ isolated from the cancer samples. With regard to *O*-glycans, type 2 *O*-glycans, in particular, were enriched at Ser127 and Ser132 in cancer and especially, in TC. The other important difference between *O*-glycosylation at Ser127 and Ser132 is a complete absence of short *O*-glycans, like Tn (*N*-acetygalactosamine attached to serine/threonine), T (galactose-β-1,3*-N*-acetygalactosamine attached to serine/threonine) and sialyl T antigen on hCGβ from TC patients in comparison with an invasive mole patient and pregnant women. The authors suggest that these glycan differences might facilitate the application of lectins to the diagnosis of malignancies (including TC), but a larger number of samples need to be analysed to confirm such results [102]. 

The study led by Lempiainen found that non-seminomatous germ cell tumours (NSGCTs) produced hCG-H when examined in tissues [103]. hCG-H was absent in seminoma, spermatocytic seminomas, pure teratomas, non-seminoma or yolk sac. This might aid in discriminating between seminoma and NSGCT with a sensitivity of 23% and specificity of 100% for NSGCT among patients with TC. The hCG-H-staining in tissues correlated very well with a serum concentration of hCG-H. In addition, the staining intensity of hCG-H correlated well with the disease stage, but did not correlate with progression-free survival [103]. Another study from the same group concluded that the measurement of hCG-H in pre-operative NSGCT and seminoma patients correlated well with hCG and hCGβ [104]. Moreover, hCG-H had the same prognostic value as hCG and hCGβ and the levels of hCG-H, hCG and hCGβ followed the disease course (during follow-up and in relapse) in the same way. Accordingly, it was concluded that hCG-H did not provide any additional clinical information over that provided by hCG and hCGβ. There are limitations in these two studies led by Lempiainen, such as the low number of samples investigated for some disease conditions, the fact that the hCG-H antibody detected hCGβ-H in only 25% of cases and that the limit of detection for hCG-H is quite high (2 pM) [104].

### 3.2. Performance of Glycosylated AFP as a TC Biomarker

It is apparent that the most appropriate lectins to interact with AFP’s glycans are Concanavalin A (Con A, a mannose binding lectin) and *Lens culinaris* agglutinin (LCA, recognising α1,6-fucose). In the early experiments, Con A affinity chromatography exhibited different binding to AFP isolated from amniotic fluid, foetal serum, liver cancer serum and yolk sac tumour serum [105]. The different fractions of AFP bind to specific lectins [106]. Studies suggested that a lectin-reactive AFP form indicated a high risk of tumour recurrence [107,108]. Fraction AFP-L3% (i.e., AFP fraction binding to *Lens culinaris* agglutinin—LCA) may be used to distinguish between benign and malignant tumours (i.e., a predictive biomarker) [109], but the same form of AFP has been produced by HCC [110]. Since AFP present in the serum of GCT patients has additional GlcNAc linked to the β-mannose core of the glycan (i.e., a bisecting glycan as shown in Figure 3a for NSGCT) patient [54], the binding of Con A is blocked. Hence, it is possible to calculate the Con A binding ratio (Con A-BR) as the percentage of AFP not bound to Con A [111].

By applying Con A-BR >15%, it was possible to distinguish patients with tumour and non-tumour liver disease from patients with GCT with a sensitivity of 98% and specificity of 98%, using a cut-off value of 15%, while the sensitivity was 100% and specificity 62% respectively, for a cut-off value of 10% [115]. In the next study by Mora’s group, 50 GCT patients with an increase of >20% in the AFP level during chemotherapy or follow-up, were investigated to determine whether elevated AFP indicated GCT progression or a hepatic disease [115]. The results exhibit a sensitivity of 96% and specificity of 0% for the measurement of the AFP level, while Con A-BR provided a sensitivity of 92% and specificity of 100% [115]. The reason why Con A-BR cannot be applied to the diagnostics of GCT is that the Con A-BR ratio was very similar for NSGCT patients (12–43%) and for patients with gastric carcinoma (18–48%), while significantly different for patients with liver disease (2–8%).

An analysis of AFP-L3%, a fraction of AFP binding to LCA, appears to be a better biomarker for identifying a recurrence of the yolk sac tumour than an analysis of AFP in serum for a neo-natal patient [116]. The 1st resection was incomplete due to a massive haemorrhage during the operation and AFP-L3% better indicated the recurrence of the disease and also successful chemotherapy and a subsequent resection than the AFP level analysis. The other very important outcome of using AFP-L3% is that neonates without any disease, but having high AFP levels, were correctly identified as healthy by measuring AFP-L3%.

The case of a 35-year-old man diagnosed with a testicular embryonal carcinoma was monitored [117]. After orchiectomy, the AFP levels were persistently elevated but a very low AFP-L3% indicated no residual or recurrent tumour. The patient has been free from disease for 3 years following the operation and the AFP levels have remained in the range of 19–27 ng/mL.

An AFP-L3% assay might provide better information about future disease recurrence/relapse for NSGCT, teratocarcinoma and embryonal cell carcinoma adult patients than the AFP level in serum, as suggested by Kawai et al. [107]. The AFP-L3% assays need to be implemented with caution as there were cases when such an assay failed to detect tumour activity and was negative in two purely seminoma patients with AFP at the cut-off value. However, the authors also conclude that AFP-L3% is a sensitive and specific marker of TC, especially in cases where the AFP level is only slightly increased or at the cut-off value of 20 ng/mL [107]. Another study indicates that AFP-L3% with a value of >50% correctly identified 96% of patients with various types of NSGCTs, i.e., embryonal carcinoma (*n* = 9), yolk sac tumour (*n* = 4), a mixed type without seminomas (*n* = 5) and a mixed type with seminoma (*n* = 7), irrespective of the AFP level in serum [109]. Moreover, in nine patients whose sera were sequentially measured, AFP-L3% was highly effective in the detection of two residual tumours, one recurrence and one false positive case [109]. 

The percentage of AFP reactive with three lectins LCA, *Phaseolus vulgaris* leukoagglutinin and *Ricinus communis* agglutinin using a radioimmunoassay showed a significant difference in the glycan composition of AFP from patients with GCT (*n* = 7) in comparison with AFP from embryonal fluids (*n* = 11) [118]. 

Johnson et al. analyzed the glycan composition of AFP isolated from the sera of 2 patients with HCC (hepatocellular carcinoma) and 2 patients with NSGCT [113] and, in a second study, from 12 HCC patients and from 1 NSGCT patient [114]. These studies identified different glycan quantities of AFP depending on the disease (see Figure 3). The authors claim that they chose samples with a high AFP level for analysis, and it is uncertain whether the differences observed in the AFP’s glycan composition isolated either from HCC or from NSCGT patients would be the same/similar if low AFP level samples were included in the studies. Thus, in the next studies, more samples need to be included and investigated. Besides having biantennary glycans on AFP as shown by Johnson [113,114], the Uniprot database (https://www.uniprot.org/uniprot/P02771) suggests the presence of triantennary and/or branched N-glycans, as was confirmed by Aoyagi et al. [112] and Tsuchida et al. [54].

### 3.3. Analysis of Glycans in Serum

Serum *N*-glycans were examined for the diagnostic and prognostic ability in GCTs. The authors performed a structural analysis of 103 age adjusted healthy volunteers and 54 patients with different types of TC. Five glycan structures were applied as diagnostics biomarkers with AUC > 0.75. The discriminative analysis of glycans as a prognostic biomarker was significant and the AUC value was 0.87, when using the *N*-glycan score (4 glycans with *m/z* 2890, 3195, 3560, and 3865). Importantly, the *N*-glycan score correctly identified 10 out of 12 (83%) patients with negative conventional tumour markers. Moreover, six different glycans were identified for prognostics (AUC = 0.89) and four glycans for relapse purposes. A survival analysis was examined during 20 months and a high value of identified glycans (mostly fucosylated and sialylated glycans) was associated with a poor prognosis. A summary of glycan structures applicable as diagnostic or prognostic biomarkers is summarised in Figure 4. The study also reached a conclusion that IgG is not a protein carrying candidate *N*-glycans, but a carrier protein was not identified [119].

### 3.4. Application of Lectins for Analysis in Tissues

Lectin histochemistry found its way for TC diagnosis and staging [120,121]. These studies have shown that different cells at different stages react with different lectins. Various types of NGCs have similar reactivity with lectins Con A, LCA, wheat germ agglutinin (WGA) and RCA I, while lectins peanut agglutinin (PNA), soybean agglutinin (SBA), *Helix pomatia* agglutinin (HPA) exhibited stronger binding towards normal cells. These findings suggested the potential use of lectins in histological identification of tumours [120,122]. 

The tissue sections from 5 different organs (brain, liver, kidney, spleen and testis) from two mice were analysed using lectin microarrays [123]. The results suggest that the overall glycome pattern of testis is completely different, when comparing with other tissues with significantly different clustering using a principal component analysis and *O*-glycosylation (Figure 5). The results also showed that the localization of *O*-glycan binders was limited at the inner part of seminiferous tubules with different staining patterns [123]. The differences in the glycan profiles of various structures within testis tissues suggests that lectins have a potential to be applied to detect malignant processes associated with TC.

When changes in the profile of simple mucin-type *O*-glycans (Tn, sialyl-Tn and T antigens) were investigated in tissues of human testis and testicular neoplasm, it was found that normal testis showed a restricted pattern when considering *O*-glycosylation and/or the expression of six polypeptide GalNAc transferases [124]. On the other hand, profound changes were associated with neoplasia in malignant germ cells [124]. 

### 3.5. Analysis of Expression Of GLycan-Processing Enzymes

An immunohistochemical analysis of the core 2 *N*-acetylglucosaminyltransferase-1 (C2GnT-1, responsible for branching of *O*-glycans) level in GCT tissues revealed a slight positivity in stage I disease (29.5%, 21/71) compared to more advanced disease stages (84.7%, 50/59) (*p* < 0.001). Moreover, C2GnT-1-positive GCT patients with stage I seminoma had a higher risk for recurrence of the disease (*p* < 0.001). The results obtained strongly suggest that C2GnT-1 enhances the metastatic potential of TGCT and may be a reliable biomarker to identify the aggressive potential of testicular GCT [125].

An immunohistochemical analysis of orchiectomy specimens of 130 patients with TGTC revealed the under-expression of *N*-acetylglucosaminyltransferase-V (GnT-V), responsible for β1-6 branching during carcinogenesis and the progression of TGCT [126]. This branching is recognised by *Phaseolus vulgaris* leukoagglutinin (PHA-L) [127]. This is why the GnT-V enzyme is considered as a promising recurrence predictor for stage I NSGCT. The *N*-linked structural analysis of glycans released from the tissues allowed the conclusion that besides GnT-V, GnT-III (responsible for production of bisecting *N*-glycans) is also downregulated. On the other hand, a structural glycan analysis revealed the up-regulation of GnT-IV enzyme and this is why the detection of these additional two enzymes can also be applied for disease monitoring [126]. 

The group led by Professor Gabius discovered that by using a set of lectins and neoglycoconjugates conjugated to bovine serum albumin, it is possible to differentiate between seminoma, embryonic carcinoma and choriocarcinoma using human tissues [128].

### 3.6. Analysis of Glycan Binding Proteins

Galectins are composed of approximately 130 amino acids with a β-galactoside-binding ability. Fifteen different types of mammalian galectins were identified. For diagnostic purposes, galectin-1 and galectin-3 are the most suitable. Galectin-1 is abundant in most organs: muscle, heart, liver, prostate, lymph nodes, spleen, thymus, placenta, testis, retina, macrophages, B-cells, T-cells, tumours. Galectin-3 is localized mainly in tumour cells, macrophages, epithelial cells, fibroblasts, activated T cells [129]. Galectin-3 is expressed in a variety of tissues and plays a role in diverse biological events, such as embryogenesis, angiogenesis, adhesion, cellular proliferation, apoptosis and the modulation of the inflammatory process and immune response. Galectin-3 has also been implicated in tumour progression and metastasis in a variety of human cancers, such as thyroid, pancreas and breast carcinomas [130]. In addition, galectin-3 expression has been reported in pig, rat and human Sertoli cells [131]. 

Three types of galectins (galectin-1, -3, -8) and anti-Ki-67, anti-bcl-2 and anti-p53 were used to measure the distance between the tumour cells and cluster radii in primary testis carcinomas lung metastases [108]. A multivariable analysis of total survival in 34 patients shows a positive correlation of the distance between the tumour cells expressing galectin-1, a negative correlation of cluster radius of tumour cells expressing galectin-3, a positive correlation of distance between lymphocytes and tumour cell expressing galectin-8 and a negative correlation between lymphocytes and tumour cell expressing a high level of galectin-8. The expression of galectin-3 binding sites in the tissues of TC patients (*n* = 34) means a better survival rate compared to tissues not expressing glycans binding galectin-3 [108].

In malignant testicular Sertoli cell tumours, the expression of galectin-3 is down-regulated while, in benign Leydig cell tumours, this expression is maintained, indicating the possible implication of this gene in the development of more aggressive testicular sex cord stromal tumours [129]. In contrast to sex cord stromal tumours, Gal-3 expression is up-regulated in testicular germ cell tumours [129].

Galectin-3 is considered a marker of aggressiveness in TC [130]. While the gene is down-regulated in normal adult testis, a higher amount was observed for seminomas GCT and much higher for non-seminomas GCT tissue. A higher galectin-3 mRNA was observed in the non-seminomas GCT cell line (NCCIT) compared to the seminoma GCT cell line (JKT1) [130].

## 4. Conclusions

The review reveals the true potential of glycan profiling to become powerful TC biomarkers besides the TC biomarkers already described and applied, such as circulating DNA, microRNA, various types of proteins and circulating tumour cells. In particular, the glycoprofiling of two TC biomarkers, like hCG and AFP, would appear to be a promising area for the discovery of novel TC biomarkers. 

With regard to changes in the glycan composition of hCG, the hyperglycosylated form of hCG or hCGβ (i.e., transformation of type 1 O-glycan into type 2 *O*-glycan on Ser132 of the β-subunit of hCG), in particular, was investigated as a promising TC biomarker for distinguishing from other types of cancer (Appendix A) for potential diagnostic purposes. Only two studies, in addition to the investigation of hCG-H or hCGβ-H, have been published to date focusing on other glycan parts of hCG. Accordingly, it can be stated that an analysis of the changes of the other three *O*-glycans (Ser121, Ser127 and Ser138) and four *N*-glycans, so far underexplored, is worth investigating to identify additional promising glycans as TC biomarkers for diagnostic and prognostic purposes and for disease recurrence evaluation. 

The glycoprofiling of AFP using two lectins, LCA and Con A, can provide useful information as to whether the elevated AFP is of testicular or hepatocellular origin. Additionally, lectin-assisted AFP assays have proven to be very useful for the monitoring of TC recurrence, the success of chemotherapy and in the identification of false positive cases. Lectins, other than these two, need to be integrated into assay formats to reveal the true potential of glycans as TC biomarkers. 

It can be expected that an analysis of glycan changes per se will not deliver reliable results for diagnostic and prognostic purposes and for recurrence evaluation, but might be a desirable solution in combination with traditional serological biomarkers as shown in Appendix A or with other novel serological biomarkers (circulating DNA, microRNA, circulating tumour cells, etc.).

## Figures and Tables

**Figure 1 diagnostics-09-00156-f001:**
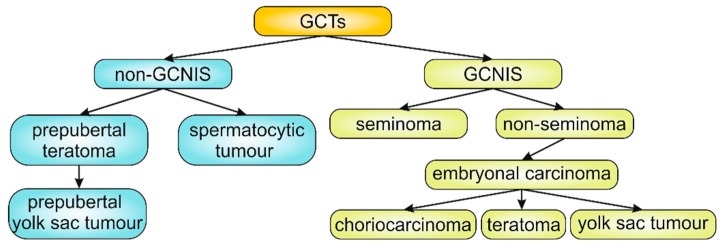
Main types of testicular cancer (TC). GCTs = germ cell tumours, GCNIS = germ cell neoplasia in situ.

**Figure 2 diagnostics-09-00156-f002:**
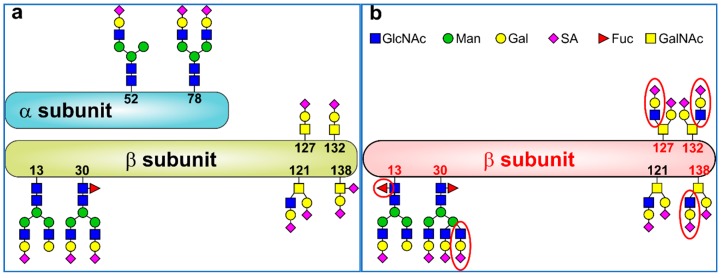
(**a**) A typical structure of hCG with 2 *N*-glycans in the α subunit of the hCG (hCGα) at Asn52 and Asn78, while the β subunit of hCG (hCGβ) contains 2 *N*-glycans (Asn13 and Asn30) and 4 *O*-glycans (Ser121, Ser127, Ser132 and Ser138). The glycan composition shown here for hCGβ is a typical structure present in the protein from a pregnant woman; (**b**) The structure of β subunit of hyperglycosylated hCG (hCG-H) i.e., hCGβ-H present in various malignancies including TC. While type 2 *O*-glycan is present in both forms of hCGβ and hCGβ-H at Ser121, hCGβ-H contains type 2 *O*-glycans at all *O*-glycan sites (Ser127, Ser132 and Ser138). The differences in the glycan composition between hCGβ and hCGβ-H are highlighted by a red ellipse. Abbreviations used: GlcNAc = *N*-acetylglucosamine, Man = mannose, Gal = galactose, SA = sialic acid, Fuc = fucose and GalNAc = *N*-acetygalactosamine. Redrawn from [35], Copyright (2016), with permission from Elsevier.

**Figure 3 diagnostics-09-00156-f003:**
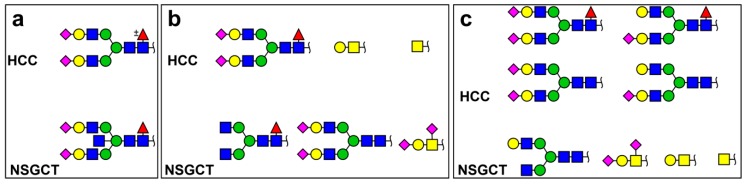
Typical glycan structures on α-fetoprotein (AFP) isolated from hepatocellular carcinoma (HCC) or NSGCT patients determined in various papers: (**a**) drawn according to information provided in ref. [112]; (**b**) Reprinted by permission from Nature, Copyright 1999 from ref. [113] and (**c**) Reprinted by permission from Nature, Copyright 2000 from ref. [114].

**Figure 4 diagnostics-09-00156-f004:**
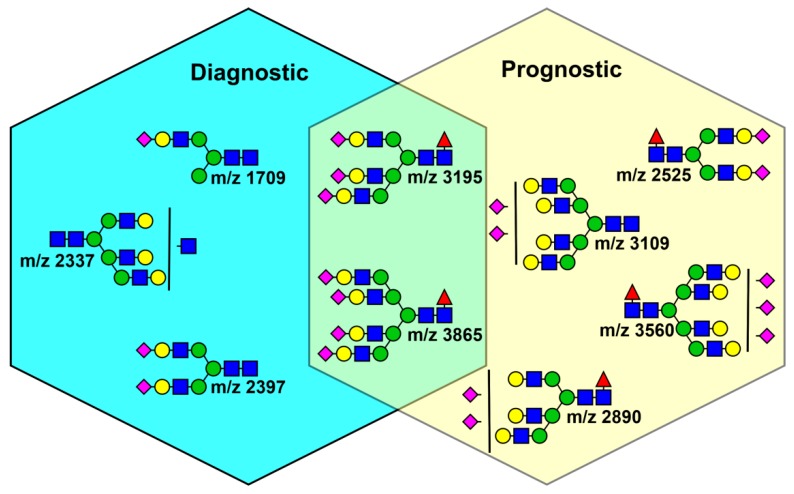
Representative diagrams of candidate *N*-glycans present in serum of GCT patients applicable as diagnostic or prognostic biomarkers. Terminal sialylated bi-antennary, tri-antennary and tetra-antennary complex-type *N*-glycans were selected as a GCT-related *N*-glycans. Reprinted from ref. [119] with modifications.

**Figure 5 diagnostics-09-00156-f005:**
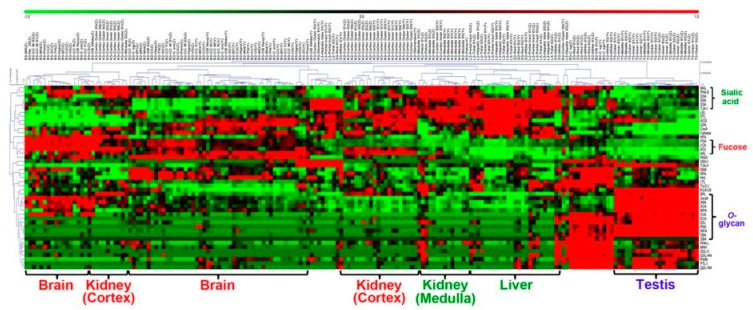
Two-dimensional analysis of the normalized lectin microarray data from the 182 tissue fragments. The 182 samples are listed in columns and the 45 lectins are listed in rows. The colour and intensity of each square indicate the lectin signal levels in specific tissue fragments (Red, high; green, low; black, medium). Reprinted from ref. [123].

**Table 1 diagnostics-09-00156-t001:** Basic characteristics of currently used TC biomarkers. Update based on Ref. [18].

TC Biomarkers	Half-Life	Normal Values	Tumour Type	Glycoprotein	Subunits
AFP	5–7 d	< 40 ng/mL	EC, T, YST	1 *N*-glycan 1 *O*-glycan	1
hCG	24–36 h	< 5 mIU/mL (0.6 ng/mL)	Seminoma, TC choriocarcinoma, EC	4 *N*-glycans 4 *O*-glycans	2
LDH	varies	1.5–3.2 nkat/mL	Any	-	4

AFP = α-fetoprotein, hCG = human chorionic gonadotropin, LDH = lactate dehydrogenase, EC = embryonal carcinoma, YST = yolk sac tumour.

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
