# Peer review of "Glycan Analysis as Biomarkers for Testicular Cancer"

_diagnostics, 2019, doi:10.3390/diagnostics9040156_

Round 1
Reviewer 1 Report
The authors described the limitation of current testicular cancer biomarkers in this review and suggested the potential of glycan changes on hCG/AFP as testicular cancer biomarkers for diagnostics and prognostics purposes.
The authors explained the description and classification of testicular cancer in the first part, but they did not collectively summarize the TC classification and the biomarkers in the next sentence. So that the focus of this review is hard to comprehend.
From the title, the most important thing in this review is to explain and demonstrate the discovery of novel clinical biomarkers clearly that may help the early detection and the monitor of the disease. By only based the limited-analysis of the glycan composition of hCG and AFP, the prospect about providing a newly glycan-based biomarkers for TC is difficult to justify. Furthermore, these studies were not proceeded or expanded upon score years ago, the glycan composition of hCG or AFP is less likely to be able used in diagnostics TC.
If the authors aimed to comprehensively review the evidence of limitation of current studies, they should modify the title and cite the newest references including original studies and the other reviews.
Author Response
Response to Reviewer´s comments
Reviewer 1
The authors described the limitation of current testicular cancer biomarkers in this review and suggested the potential of glycan changes on hCG/AFP as testicular cancer biomarkers for diagnostics and prognostics purposes.
R: In the revised version of the manuscript we extended the scope of the review paper by adding four new sub-chapters relevant to glycan analysis: 3.3. Analysis of glycans in serum; 3.4. Application of lectins for analysis in tissues; 3.5. Analysis of expression of glycan-processing enzymes; 3.6. Analysis of glycan binding proteins with the following text:
3.3. Analysis of glycans in serum
Serum N-glycans were examined for the diagnostic and prognostic ability in GCTs. Authors performed structural analysis of 103 age adjusted healthy volunteers and 54 patients with different types of TC. Five glycan structures were applied as diagnostics biomarkers with AUC >0.75. The discriminative analysis of glycans as prognostic biomarker was significant, and the AUC value was 0.87, when using the N-glycan score (4 glycans with m/z 2890, 3195, 3560, and 3865). Importantly the N-glycan score correctly identified 10 out of 12 (83%) of patients with negative conventional tumour markers. Moreover, six different glycans were identified for prognostics (AUC=0.89) and four glycans for relapse purposes. Survival analysis was examined during 20 months and high value of identified glycans (mostly fucosylated and sialylated glycans) was associated with a poor prognosis. A summary of glycan structures applicable as diagnostic or prognostic biomarkers is summarised in Fig. 4. The study also reached a conclusion that IgG is not a protein carrying candidate N-glycans, but a carrier protein was not identified [119].
3.4. Application of lectins for analysis in tissues
Lectin histochemistry found its way for TC diagnosis and staging [120, 121]. These studies have shown that different cells at different stages react with different lectins. Various types of NGCs have similar reactivity with lectins Con A, LCA, wheat germ agglutinin (WGA) and RCA I, while lectins peanut agglutinin (PNA), soybean agglutinin (SBA), Helix pomatia agglutinin (HPA) exhibited stronger binding towards normal cells. These findings suggested potential use of lectins in histological identification of tumours [120, 122].
Tissue sections from 5 different organs (brain, liver, kidney, spleen and testis) from two mice was analysed using lectin microarrays [123]. The results suggest that the overall glycome pattern of testis is completely different, when comparing with other tissues with significantly different clustering using principal component analysis and O-glycosylation (Fig. 5). The results also showed that the localization of O-glycan binders was limited at the inner part of seminiferous tubules with different staining patterns [123]. Differences in glycan profiles of various structures within testis tissues suggests that lectins have a potential to be applied to detect malignant processes associated with TC.
When changes in the profile of simple mucin-type O-glycans (Tn, sialyl-Tn and T antigens) was investigated in tissues of human testis and testicular neoplasm, it was found out that normal testis showed a restricted pattern, when considering O-glycosylation and/or expression of six polypeptide GalNAc transferases [124]. On the other hand profound changes were associated with neoplasia in malignant germ cells [124].
3.5. Analysis of expression of glycan-processing enzymes
Immunohistochemical analysis of core 2 N-acetylglucosaminyltransferase-1 (C2GnT-1, responsible for branching of O-glycans) level in GCT tissues revealed a slight positivity in stage I disease (29.5%, 21/71) compared to more advanced disease stages (84.7%, 50/59) (P < 0.001). Moreover, C2GnT-1-positive GCT patients with stage I seminoma had a higher risk for recurrence of the disease (P < 0.001). The results obtained strongly suggest that C2GnT-1 enhances the metastatic potential of TGCT and may be a reliable biomarker to identify aggressive potential of testicular GCT [125].
Immunohistochemical analysis of orchiectomy specimens of 130 patients with TGTC revealed under-expression of N-acetylglucosaminyltransferase-V (GnT-V), responsible for 1-6 branching during carcinogenesis and progression of TGCT [126]. Such branching is recognised by Phaseolus vulgaris leukoagglutinin (PHA-L) [127]. This is why GnT-V enzyme is considered as a promising recurrence predictor for stage I NSGCT. N-linked structural analysis of glycans released from the tissues allowed concluding that besides GnT-V, also GnT-III (responsible for production of bisecting N-glycans) is also downregulated. On the other hand, structural glycan analysis revealed up-regulation of GnT-IV enzyme and this is why detection of these additional two enzymes can be applied for disease monitoring, as well [126].
The group led by Prof. Gabius discovered that by using set of lectins and neoglycoconjugates conjugated to bovine serum albumin it is possible to differentiate between seminoma, embryonic carcinoma and choriocarcinoma using human tissues [128].
3.6. Analysis of glycan binding proteins
Galectins are composed of about 130 amino acids with a β-galactoside-binding ability. Fifteen different types of mammalian galectins were identified. For diagnostic purposes especially galectin-1 and galectin-3 are the most suitable. Galectin-1 is abundant in most organs: muscle, heart, liver, prostate, lymph nodes, spleen, thymus, placenta, testis, retina, macrophages, B-cells, T-cells, tumours. Galectin-3 is localized in mainly in tumour cells, macrophages, epithelial cells, fibroblasts, activated T cells [129]. Galectin-3 is expressed in a variety of tissues and plays a role in diverse biological events, such as embryogenesis, angiogenesis, adhesion, cellular proliferation, apoptosis, and modulation of the inflammatory process and immune response. Galectin-3 has also been implicated in tumour progression and metastasis in a variety of human cancers such as thyroid, pancreas and breast carcinomas [130]. In addition, galectin-3 expression has been reported in pig, rat and human Sertoli cells [131].
Three types of galectins (galectin-1,-3,-8) and anti-Ki-67, anti-bcl-2 and anti-p53 were used to measure the distance between the tumour cells and cluster radii in primary testis carcinomas lung metastases [108]. Multivariable analysis of total survival in 34 patients shows a positive correlation of distance between tumour cells expressing galectin-1, a negative correlation of cluster radius of tumour cells expressing galectin-3, a positive correlation of distance between lymphocytes and tumour cell expressing galectin-8 and a negative correlation between lymphocytes and tumour cell expressing high level of galectin-8. Expression of galectin-3 binding sites in the tissues of TC patients (n=34) means a better survival rate compared to tissues not expressing glycans binding galectin-3 [108].
In malignant testicular Sertoli cell tumours, the expression of galectin-3 is down-regulated while, in benign Leydig cell tumours, this expression is maintained, indicating possible implication of this gene in the development of more aggressive testicular sex cord stromal tumours [129]. In contrast to sex cord stromal tumours, Gal-3 expression is up-regulated in testicular germ cell tumours [129]
Galectin-3 is considered a marker of aggressiveness in TC [130]. While the gene is down-regulated in normal adult testis, higher amount was observed for seminomas GCT and much higher for non-seminomas GCT tissue. A higher galectin-3 mRNA was observed in non-seminomas GCT cell line (NCCIT) compared to seminoma GCT cell line (JKT1) [130].”
R: Moreover the part dealing with hCG properties was moved from ESI file into the main text of the manuscript and can be read as follows:
“Hyperglycosylated forms of HCG are currently defined as drivers for most or possibly all human cancers [46]. The protein hCG and its various forms can be divided into several groups:
two types when considering binding preference [41]: binding to LH/hCG hormone receptor (hormone hCG, a normal form of hCG); binding to TGFb-II receptor (hCG-H or hCGb-H).
four molecules, when considering glycosylation pattern containing [47, 48]: four biantennary N-glycans, three type 1 O-glycans (tri- and tetra-saccharides), one type 2 O-glycan (hCG); four biantennary N-glycans and four Type 2 O-glycans (penta- and hexa-saccharides) (placental hyperglycosylated hCG); triantennary N-glycans on β-subunit and Type 2 O-glycans (extravillous cytotrophoblast hyperglycosylated hCG and cancer hyperglycosylated hCG); N-acetylgalactosamine-sulphate terminating N-glycans and type 1 O-glycans (sulphated hCG hormone).
seven semi-independent molecules, when considering functional roles [41]: placental hCG (controlling pregnancy); placental autocrine hyperglycosylated hCG (start placentation during pregnancy); pituitary sulphated hCG (present during ovarian steroidogenesis, ovulation & luteogenesis); fetal hCG hormone (promoting fetal organ growth); ovarian hyperglycosylated hCG (driving the final proteolytic enzymatic step during ovulation); hCG-H (trophoblastic tumours); hCGb-H (non-trophoblastic tumours).
significant structural variability involving numerous isoforms combining glycosylation (14 major variants for hCGa and 12 major variants for hCGb) [35, 49, 50] and structural (nicked forms of hCG, hCGb, hCG-H, hCGb-H; hCG forms without a terminal CTP tail (b113-145 containing all 4 O-glycans); b-core fragment (b6-40 and b55-92 linked by a disulfide bond)) variations [35, 41].”
The authors explained the description and classification of testicular cancer in the first part, but they did not collectively summarize the TC classification and the biomarkers in the next sentence. So that the focus of this review is hard to comprehend.
R: The initial general part of the review was extended and divided into three sub-chapters: 1.1. TC risk factors and 1.2. TC classification with the following text:
TC risk factors
Recent studies suggest that TC is a result of interactions of multiple factors including environmental and genetic ones, a claim supported by the study involving monozygotic/dizygotic twins [4, 5]. Disrupted foetal hormone signalling during foetal development due to the influence of various xenobiotics as well as misbalanced maternal hormone levels is thought to be the cause of multiple disorders linked to TC [6-9]. A congenital defect, when one or both testicles are undescended into the scrotum (cryptorchidism), is connected with 5-fold increase of TC risk [3, 10, 11]. Furthermore, inflammation of the testicles, for example, due to overcoming mumps and various testicular injuries has a negative effect, as well [6]. Positive family history might be a strong risk factor since the risk of TC development increases, when TC is diagnosed for direct blood relative men like father and brother [12, 13]. When TC was diagnosed in one testis, there is a risk of disease developing in the opposite testis. Environmental impact on TC development and progression is supported by the study concluding that sons of men who moved from a low- to a high-incidence place have the same risk of TC as males living at a high-incidence place and the effect of harmful environmental factors can be observed already in the 2nd generation [14]. Some information indicate that postnatal environmental and lifestyle (a diet and an exposure to endocrine disrupting agents) factors may negatively influence development of TC [15].
TC classification
Rapid and efficient diagnosis, as well as the prognosis, is highly complicated due to high diversity of the disease. The most common first symptoms are change in the area of the testicles. Gradually growing lump of solid consistency with a smooth, sometimes bumpy surface are detected by palpation randomly or during examination. Only 20% of patients suffer from pain, which is described like a blunt pain and only accidentally as a severe pain with a risk of tumour bleeding. Much more is discovered by physicians in the investigation of patients who suffer from testis and prostate atrophy, feminization, gynecomastia, hair loss or libido change due to hormonal changes. TC is classified according to the histological tissue composition, the germ cell lineage and age at onset of TC [3, 16]. More than 90% - 95 % of TC are germ cell tumours (GCTs) affecting testicular germ cells (cells making sperms) [6, 17]. The current WHO classification system defines two major entities of GCTs as germ cell neoplasia in situ (GCNIS)-related and non-GCNIS-related (non-GCNIS) (Fig. 1) [3, 16]. Approximately 60% of GCTs contain more than one type of histological pattern (i.e. mixed GCTs) [18].
The remaining TCs are of non-germ cell origin and include sex cord and gonadal stromal tumours, lymphoid and hematopoietic tumours, and metastatic tumours from other primary neoplasms [17, 19, 20]. From pure non-germ cell tumours, the largest occurrence has Leydig cell tumour (LCT), then Sertoli cell tumour (SCT), granulosa cell tumour, and pure stromal tumour [19, 21].”
From the title, the most important thing in this review is to explain and demonstrate the discovery of novel clinical biomarkers clearly that may help the early detection and the monitor of the disease. By only based the limited-analysis of the glycan composition of hCG and AFP, the prospect about providing a newly glycan-based biomarkers for TC is difficult to justify. Furthermore, these studies were not proceeded or expanded upon score years ago, the glycan composition of hCG or AFP is less likely to be able used in diagnostics TC.
R: New paragraphs under sub-chapters 2.3. Lactate dehydrogenase (LDH) and 2.4. Other TC biomarkers were added with the following text:
“2.3. Lactate dehydrogenase (LDH)
Lactate dehydrogenase (LDH; 134 kDa) is a cytoplasmic enzyme produced by many types of tissues including muscle (skeletal, smooth, cardiac), liver, kidney and brain [18, 57, 58]. LDH is a tetrameric protein composed of two structurally different subunits. In serum, the enzyme is present in a form of 5 isoenzymes, which concentration correlates with number of gene copies in a short arm of chromosome 12p, where they are coded [18, 37, 59]. Isoenzyme LDH-1 is the most frequent form of LDH present during increased levels of LDH [18, 37, 60]. Although 40-60% of men with TC of germ cell etymology have elevated LDH, this marker may be considered alternative serological markers due to relatively low specificity for GCTs compared to AFP and hCG [18, 37, 58, 61]. hCG and AFP are produced by tumour cells, but increased level of LDH in serum is a result of a cell damage, as well [57]. Nonetheless, there are some correlation between LDH levels and survival of TC patients [37, 62]. LDH is one of the three sole serum markers currently used for risk stratification in TC, based on International Germ Cell Consensus Classification (IGCCC) criteria [32, 63]. LDH has a limited sensitivity and specificity for seminoma, it is increased in about 80% of advanced seminomas. Elevated LDH level was observed in 60% of advanced non-seminomas with a value higher than 2000 U/L, indicating an advanced disease and/or disease recurrence [18, 58, 64]. LDH assays determine activity not quantity, therefore differences in values can be expected between several methods applied for its activity assays [18].
2.4. Other TC biomarkers
A recent review made the following conclusion: “Having systematically reviewed the available literature, we found surprisingly little evidence to guide optimal testing with biomarkers (AFP, hCG and LDH) routinely used during follow-up for testicular cancer recurrence“ [65]. Due to all these reasons there is still a substantial effort to find novel biomarkers which will be more reliable for example for disease prognosis [66] or diagnostics (DNA methylation, microRNAs, proteins) [17, 67-70].
Neuron-specific enolase, an isoenzyme of the glycolytic enzyme 2-phospho-D-glycerate-hydrolase, is elevated in about 30 – 50% of patients with seminoma, specifically in metastatic stages. Moreover the protein level can be increased in patients with normal hCG and LDH concentration [17, 71, 72]. On the other hand, it is not a reliable marker due to high false-positive rate [73]. Elevation was observed also in other conditions [24, 74].
Another TC biomarker is placental alkaline phosphatase. There are 2 genes coding the proteins with alkaline phosphatase activity, the placental (PLAP) and germ cell enzyme (also noted placental-like alkaline phosphatase) [75]. The enzyme is physiologically expressed in foetal germ cells and in infants [18]. Therefore, the staining results in the first years of postnatal life must be interpreted with caution [25]. The protein is also produced ectopically by a variety of malignant tumours [76, 77]. Elevated protein level is observed in about 80% of TC patients [78] and the most frequently in seminoma TC (60-70%) [79]. Despite the low false-positive rates (1.6%), its potential for disease monitoring is complicated by the fact that its serum level can increase up to 10-fold by smoking [26, 65, 78, 79].
TC cells express several high molecular weight glycoproteins. One of this antigen, sialylated keratin sulphate proteoglycan, can be detected by monoclonal antibody against TRA-1-60 (podocalyxin) [80, 81]. It is expressed by embryonal cancer, seminoma and carcinoma in situ of the testis [82, 83]. The study showed that the antigen is expressed in approximately 80% of patients with advanced embryonal carcinoma. Although its level decreases during chemotherapy, 15 - 30% patients do not have normalized level after therapy [84]. Low assay specificity limits its wider use [17, 18, 37].
From novel TC biomarkers discovered so far we can list the following biomolecules/cells: microRNAs [26, 85], DNA methylation [86], circulating tumour cells [26], circulating DNA [26], various proteins [26]. DNA-based biomarkers can be used for non-invasive diagnostics due to presence of different DNA types in blood stream [87]. It was observed an increased level of circulating tumour DNA in men with TC and circulating tumour DNA can distinguish patients with cancer from healthy ones (88% sensitivity and 97% specificity), also in cases with normal level of conventional markers [88, 89]. DNA present in blood is produced by different organs/cells and this is why by DNA analysis we can get clearer and more complex picture about substantial heterogeneity of TC. On the other hand, DNA is unstable in blood stream and it is rapidly cleared. Therefore use of this marker for diagnosis requires high-throughput and sensitive techniques [17]. The largest potential from novel TC biomarkers have microRNAs associated with different types of TC [68]. MicroRNAs are highly stable versions of RNA modulating protein-coding genes expression. MicroRNAs act either as oncogenes or tumour‑suppressor genes. In cancer they are dysregulated, and their profiles can show the origin of tumours. Thank to these properties, microRNAs are promising biomarkers for cancer monitoring [32]. For example TC patients negative for microRNA‐371a‐3p had a better progression-free survival and an overall survival compared to the TC patients with microRNA‐371a‐3p present in serum [90].
None of the mentioned markers are universal and specific. Recently it was discovered that extracellular vesicles such as exosomes can be a rich source of various types of biomarkers, as detected for various types of urological tumours [91]. Since so far exosomes have not been applied for TC diagnostics and/or monitoring their application in discovery for novel and robust TC biomarkers is extremely exciting.”
If the authors aimed to comprehensively review the evidence of limitation of current studies, they should modify the title and cite the newest references including original studies and the other reviews.
R: The title of the review paper was modified to “Glycan analysis as biomarkers for testicular cancer” with significant number of new references including reviews added. Moreover we added two new figures into the revised version of the manuscript. Fig. 4 was redrawn using info provided in ref. 119 and Fig. 5 was taken from ref. 123. Please see the file “Revised manuscript with changes highlighted.docx” showing all changes made in the revised version of the manuscript.

Reviewer 2 Report
This review article pointed out that currently two of the most important testicular cancer (TC) biomarkers, a-fetoprotein (AFP) and human chorionic gonadotropin (hCG), are not produced solely due to TC and up to 40% of patients with TC do not have elevated serological biomarkers. To find a new biomarker, this article firstly showed protein glycoprofiling as a promising biomarker for TC disease diagnostics, monitoring and recurrence evaluation.
It is straightforward, well written, and concise and has clear results. It definitely deserves to be published and is a valuable contribution to the Diagnostics. Some minor issues could be addressed before publication.
With just a few exceptions, the paper is well design and obtain a good diagnostic value for TC.
Minor points:
If the author writes this article, following the three parts in the abstract, for example, ‘1. Description and classification of TC’, ‘2. The limitations of current TC biomarkers’ and ‘3. The new TC biomarkers for diagnostics and prognostics purposes and for disease recurrence evaluation’. Then put the subtitles ‘Testicular cancer (TC)’ et al under these three major titles, which would be perfect.
Author Response
Response to Reviewer´s comments
Reviewer 2
This review article pointed out that currently two of the most important testicular cancer (TC) biomarkers, a-fetoprotein (AFP) and human chorionic gonadotropin (hCG), are not produced solely due to TC and up to 40% of patients with TC do not have elevated serological biomarkers. To find a new biomarker, this article firstly showed protein glycoprofiling as a promising biomarker for TC disease diagnostics, monitoring and recurrence evaluation.
It is straightforward, well written, and concise and has clear results. It definitely deserves to be published and is a valuable contribution to the Diagnostics. Some minor issues could be addressed before publication.
With just a few exceptions, the paper is well design and obtain a good diagnostic value for TC.
Minor points:
If the author writes this article, following the three parts in the abstract, for example, ‘1. Description and classification of TC’, ‘2. The limitations of current TC biomarkers’ and ‘3. The new TC biomarkers for diagnostics and prognostics purposes and for disease recurrence evaluation’. Then put the subtitles ‘Testicular cancer (TC)’ et al under these three major titles, which would be perfect.
R: Initial part of the review paper was divided into two parts:
“1. Description, risk factors and classification of TC
Cancer is either the first or second most common cause of death below the age of 70 in more than half of 172 countries, according to the estimates from the World Health Organisation in 2015 [1]. It is predicted that 9.6 million people died as a result of cancer and 18.1 million new cases were recorded worldwide in 2018 [1]. Testicular cancer (TC) is relatively rare, but it is the most common solid tumour found in young men [2, 3].
TC risk factors
Recent studies suggest that TC is a result of interactions of multiple factors including environmental and genetic ones, a claim supported by the study involving monozygotic/dizygotic twins [4, 5]. Disrupted foetal hormone signalling during foetal development due to the influence of various xenobiotics as well as misbalanced maternal hormone levels is thought to be the cause of multiple disorders linked to TC [6-9]. A congenital defect, when one or both testicles are undescended into the scrotum (cryptorchidism), is connected with 5-fold increase of TC risk [3, 10, 11]. Furthermore, inflammation of the testicles, for example, due to overcoming mumps and various testicular injuries has a negative effect, as well [6]. Positive family history might be a strong risk factor since the risk of TC development increases, when TC is diagnosed for direct blood relative men like father and brother [12, 13]. When TC was diagnosed in one testis, there is a risk of disease developing in the opposite testis. Environmental impact on TC development and progression is supported by the study concluding that sons of men who moved from a low- to a high-incidence place have the same risk of TC as males living at a high-incidence place and the effect of harmful environmental factors can be observed already in the 2nd generation [14]. Some information indicate that postnatal environmental and lifestyle (a diet and an exposure to endocrine disrupting agents) factors may negatively influence development of TC [15].
TC classification
Rapid and efficient diagnosis, as well as the prognosis, is highly complicated due to high diversity of the disease. The most common first symptoms are change in the area of the testicles. Gradually growing lump of solid consistency with a smooth, sometimes bumpy surface are detected by palpation randomly or during examination. Only 20% of patients suffer from pain, which is described like a blunt pain and only accidentally as a severe pain with a risk of tumour bleeding. Much more is discovered by physicians in the investigation of patients who suffer from testis and prostate atrophy, feminization, gynecomastia, hair loss or libido change due to hormonal changes. TC is classified according to the histological tissue composition, the germ cell lineage and age at onset of TC [3, 16]. More than 90% - 95 % of TC are germ cell tumours (GCTs) affecting testicular germ cells (cells making sperms) [6, 17]. The current WHO classification system defines two major entities of GCTs as germ cell neoplasia in situ (GCNIS)-related and non-GCNIS-related (non-GCNIS) (Fig. 1) [3, 16]. Approximately 60% of GCTs contain more than one type of histological pattern (i.e. mixed GCTs) [18].
The remaining TCs are of non-germ cell origin and include sex cord and gonadal stromal tumours, lymphoid and hematopoietic tumours, and metastatic tumours from other primary neoplasms [17, 19, 20]. From pure non-germ cell tumours, the largest occurrence has Leydig cell tumour (LCT), then Sertoli cell tumour (SCT), granulosa cell tumour, and pure stromal tumour [19, 21].”
The part dealing with TC biomarkers and their limitations was extended by adding sub-chapter 2.3. Lactate dehydrogenase (LDH) with the following text:
“2.3. Lactate dehydrogenase (LDH)
Lactate dehydrogenase (LDH; 134 kDa) is a cytoplasmic enzyme produced by many types of tissues including muscle (skeletal, smooth, cardiac), liver, kidney and brain [18, 57, 58]. LDH is a tetrameric protein composed of two structurally different subunits. In serum, the enzyme is present in a form of 5 isoenzymes, which concentration correlates with number of gene copies in a short arm of chromosome 12p, where they are coded [18, 37, 59]. Isoenzyme LDH-1 is the most frequent form of LDH present during increased levels of LDH [18, 37, 60]. Although 40-60% of men with TC of germ cell etymology have elevated LDH, this marker may be considered alternative serological markers due to relatively low specificity for GCTs compared to AFP and hCG [18, 37, 58, 61]. hCG and AFP are produced by tumour cells, but increased level of LDH in serum is a result of a cell damage, as well [57]. Nonetheless, there are some correlation between LDH levels and survival of TC patients [37, 62]. LDH is one of the three sole serum markers currently used for risk stratification in TC, based on International Germ Cell Consensus Classification (IGCCC) criteria [32, 63]. LDH has a limited sensitivity and specificity for seminoma, it is increased in about 80% of advanced seminomas. Elevated LDH level was observed in 60% of advanced non-seminomas with a value higher than 2000 U/L, indicating an advanced disease and/or disease recurrence [18, 58, 64]. LDH assays determine activity not quantity, therefore differences in values can be expected between several methods applied for its activity assays [18].”
The part dealing with hCG was updated with the following text:
“Hyperglycosylated forms of HCG are currently defined as drivers for most or possibly all human cancers [46]. The protein hCG and its various forms can be divided into several groups:
two types when considering binding preference [41]: binding to LH/hCG hormone receptor (hormone hCG, a normal form of hCG); binding to TGFb-II receptor (hCG-H or hCGb-H).
four molecules, when considering glycosylation pattern containing [47, 48]: four biantennary N-glycans, three type 1 O-glycans (tri- and tetra-saccharides), one type 2 O-glycan (hCG); four biantennary N-glycans and four Type 2 O-glycans (penta- and hexa-saccharides) (placental hyperglycosylated hCG); triantennary N-glycans on β-subunit and Type 2 O-glycans (extravillous cytotrophoblast hyperglycosylated hCG and cancer hyperglycosylated hCG); N-acetylgalactosamine-sulphate terminating N-glycans and type 1 O-glycans (sulphated hCG hormone).
seven semi-independent molecules, when considering functional roles [41]: placental hCG (controlling pregnancy); placental autocrine hyperglycosylated hCG (start placentation during pregnancy); pituitary sulphated hCG (present during ovarian steroidogenesis, ovulation & luteogenesis); fetal hCG hormone (promoting fetal organ growth); ovarian hyperglycosylated hCG (driving the final proteolytic enzymatic step during ovulation); hCG-H (trophoblastic tumours); hCGb-H (non-trophoblastic tumours).
significant structural variability involving numerous isoforms combining glycosylation (14 major variants for hCGa and 12 major variants for hCGb) [35, 49, 50] and structural (nicked forms of hCG, hCGb, hCG-H, hCGb-H; hCG forms without a terminal CTP tail (b113-145 containing all 4 O-glycans); b-core fragment (b6-40 and b55-92 linked by a disulfide bond)) variations [35, 41].”
A new sub-chapter dealing with new TC biomarkers was added with the following text:
“2.4. Other TC biomarkers
A recent review made the following conclusion: “Having systematically reviewed the available literature, we found surprisingly little evidence to guide optimal testing with biomarkers (AFP, hCG and LDH) routinely used during follow-up for testicular cancer recurrence“ [65]. Due to all these reasons there is still a substantial effort to find novel biomarkers which will be more reliable for example for disease prognosis [66] or diagnostics (DNA methylation, microRNAs, proteins) [17, 67-70].
Neuron-specific enolase, an isoenzyme of the glycolytic enzyme 2-phospho-D-glycerate-hydrolase, is elevated in about 30 – 50% of patients with seminoma, specifically in metastatic stages. Moreover the protein level can be increased in patients with normal hCG and LDH concentration [17, 71, 72]. On the other hand, it is not a reliable marker due to high false-positive rate [73]. Elevation was observed also in other conditions [24, 74].
Another TC biomarker is placental alkaline phosphatase. There are 2 genes coding the proteins with alkaline phosphatase activity, the placental (PLAP) and germ cell enzyme (also noted placental-like alkaline phosphatase) [75]. The enzyme is physiologically expressed in foetal germ cells and in infants [18]. Therefore, the staining results in the first years of postnatal life must be interpreted with caution [25]. The protein is also produced ectopically by a variety of malignant tumours [76, 77]. Elevated protein level is observed in about 80% of TC patients [78] and the most frequently in seminoma TC (60-70%) [79]. Despite the low false-positive rates (1.6%), its potential for disease monitoring is complicated by the fact that its serum level can increase up to 10-fold by smoking [26, 65, 78, 79].
TC cells express several high molecular weight glycoproteins. One of this antigen, sialylated keratin sulphate proteoglycan, can be detected by monoclonal antibody against TRA-1-60 (podocalyxin) [80, 81]. It is expressed by embryonal cancer, seminoma and carcinoma in situ of the testis [82, 83]. The study showed that the antigen is expressed in approximately 80% of patients with advanced embryonal carcinoma. Although its level decreases during chemotherapy, 15 - 30% patients do not have normalized level after therapy [84]. Low assay specificity limits its wider use [17, 18, 37].
From novel TC biomarkers discovered so far we can list the following biomolecules/cells: microRNAs [26, 85], DNA methylation [86], circulating tumour cells [26], circulating DNA [26], various proteins [26]. DNA-based biomarkers can be used for non-invasive diagnostics due to presence of different DNA types in blood stream [87]. It was observed an increased level of circulating tumour DNA in men with TC and circulating tumour DNA can distinguish patients with cancer from healthy ones (88% sensitivity and 97% specificity), also in cases with normal level of conventional markers [88, 89]. DNA present in blood is produced by different organs/cells and this is why by DNA analysis we can get clearer and more complex picture about substantial heterogeneity of TC. On the other hand, DNA is unstable in blood stream and it is rapidly cleared. Therefore use of this marker for diagnosis requires high-throughput and sensitive techniques [17]. The largest potential from novel TC biomarkers have microRNAs associated with different types of TC [68]. MicroRNAs are highly stable versions of RNA modulating protein-coding genes expression. MicroRNAs act either as oncogenes or tumour‑suppressor genes. In cancer they are dysregulated, and their profiles can show the origin of tumours. Thank to these properties, microRNAs are promising biomarkers for cancer monitoring [32]. For example TC patients negative for microRNA‐371a‐3p had a better progression-free survival and an overall survival compared to the TC patients with microRNA‐371a‐3p present in serum [90].
None of the mentioned markers are universal and specific. Recently it was discovered that extracellular vesicles such as exosomes can be a rich source of various types of biomarkers, as detected for various types of urological tumours [91]. Since so far exosomes have not been applied for TC diagnostics and/or monitoring their application in discovery for novel and robust TC biomarkers is extremely exciting.”
Finally, the part dealing with glycan analysis as TC biomarkers was completed by adding four sub-chapters with the following text:
3.3. Analysis of glycans in serum
Serum N-glycans were examined for the diagnostic and prognostic ability in GCTs. Authors performed structural analysis of 103 age adjusted healthy volunteers and 54 patients with different types of TC. Five glycan structures were applied as diagnostics biomarkers with AUC >0.75. The discriminative analysis of glycans as prognostic biomarker was significant, and the AUC value was 0.87, when using the N-glycan score (4 glycans with m/z 2890, 3195, 3560, and 3865). Importantly the N-glycan score correctly identified 10 out of 12 (83%) of patients with negative conventional tumour markers. Moreover, six different glycans were identified for prognostics (AUC=0.89) and four glycans for relapse purposes. Survival analysis was examined during 20 months and high value of identified glycans (mostly fucosylated and sialylated glycans) was associated with a poor prognosis. A summary of glycan structures applicable as diagnostic or prognostic biomarkers is summarised in Fig. 4. The study also reached a conclusion that IgG is not a protein carrying candidate N-glycans, but a carrier protein was not identified [119].
3.4. Application of lectins for analysis in tissues
Lectin histochemistry found its way for TC diagnosis and staging [120, 121]. These studies have shown that different cells at different stages react with different lectins. Various types of NGCs have similar reactivity with lectins Con A, LCA, wheat germ agglutinin (WGA) and RCA I, while lectins peanut agglutinin (PNA), soybean agglutinin (SBA), Helix pomatia agglutinin (HPA) exhibited stronger binding towards normal cells. These findings suggested potential use of lectins in histological identification of tumours [120, 122].
Tissue sections from 5 different organs (brain, liver, kidney, spleen and testis) from two mice was analysed using lectin microarrays [123]. The results suggest that the overall glycome pattern of testis is completely different, when comparing with other tissues with significantly different clustering using principal component analysis and O-glycosylation (Fig. 5). The results also showed that the localization of O-glycan binders was limited at the inner part of seminiferous tubules with different staining patterns [123]. Differences in glycan profiles of various structures within testis tissues suggests that lectins have a potential to be applied to detect malignant processes associated with TC.
When changes in the profile of simple mucin-type O-glycans (Tn, sialyl-Tn and T antigens) was investigated in tissues of human testis and testicular neoplasm, it was found out that normal testis showed a restricted pattern, when considering O-glycosylation and/or expression of six polypeptide GalNAc transferases [124]. On the other hand profound changes were associated with neoplasia in malignant germ cells [124].
3.5. Analysis of expression of glycan-processing enzymes
Immunohistochemical analysis of core 2 N-acetylglucosaminyltransferase-1 (C2GnT-1, responsible for branching of O-glycans) level in GCT tissues revealed a slight positivity in stage I disease (29.5%, 21/71) compared to more advanced disease stages (84.7%, 50/59) (P < 0.001). Moreover, C2GnT-1-positive GCT patients with stage I seminoma had a higher risk for recurrence of the disease (P < 0.001). The results obtained strongly suggest that C2GnT-1 enhances the metastatic potential of TGCT and may be a reliable biomarker to identify aggressive potential of testicular GCT [125].
Immunohistochemical analysis of orchiectomy specimens of 130 patients with TGTC revealed under-expression of N-acetylglucosaminyltransferase-V (GnT-V), responsible for 1-6 branching during carcinogenesis and progression of TGCT [126]. Such branching is recognised by Phaseolus vulgaris leukoagglutinin (PHA-L) [127]. This is why GnT-V enzyme is considered as a promising recurrence predictor for stage I NSGCT. N-linked structural analysis of glycans released from the tissues allowed concluding that besides GnT-V, also GnT-III (responsible for production of bisecting N-glycans) is also downregulated. On the other hand, structural glycan analysis revealed up-regulation of GnT-IV enzyme and this is why detection of these additional two enzymes can be applied for disease monitoring, as well [126].
The group led by Prof. Gabius discovered that by using set of lectins and neoglycoconjugates conjugated to bovine serum albumin it is possible to differentiate between seminoma, embryonic carcinoma and choriocarcinoma using human tissues [128].
3.6. Analysis of glycan binding proteins
Galectins are composed of about 130 amino acids with a β-galactoside-binding ability. Fifteen different types of mammalian galectins were identified. For diagnostic purposes especially galectin-1 and galectin-3 are the most suitable. Galectin-1 is abundant in most organs: muscle, heart, liver, prostate, lymph nodes, spleen, thymus, placenta, testis, retina, macrophages, B-cells, T-cells, tumours. Galectin-3 is localized in mainly in tumour cells, macrophages, epithelial cells, fibroblasts, activated T cells [129]. Galectin-3 is expressed in a variety of tissues and plays a role in diverse biological events, such as embryogenesis, angiogenesis, adhesion, cellular proliferation, apoptosis, and modulation of the inflammatory process and immune response. Galectin-3 has also been implicated in tumour progression and metastasis in a variety of human cancers such as thyroid, pancreas and breast carcinomas [130]. In addition, galectin-3 expression has been reported in pig, rat and human Sertoli cells [131].
Three types of galectins (galectin-1,-3,-8) and anti-Ki-67, anti-bcl-2 and anti-p53 were used to measure the distance between the tumour cells and cluster radii in primary testis carcinomas lung metastases [108]. Multivariable analysis of total survival in 34 patients shows a positive correlation of distance between tumour cells expressing galectin-1, a negative correlation of cluster radius of tumour cells expressing galectin-3, a positive correlation of distance between lymphocytes and tumour cell expressing galectin-8 and a negative correlation between lymphocytes and tumour cell expressing high level of galectin-8. Expression of galectin-3 binding sites in the tissues of TC patients (n=34) means a better survival rate compared to tissues not expressing glycans binding galectin-3 [108].
In malignant testicular Sertoli cell tumours, the expression of galectin-3 is down-regulated while, in benign Leydig cell tumours, this expression is maintained, indicating possible implication of this gene in the development of more aggressive testicular sex cord stromal tumours [129]. In contrast to sex cord stromal tumours, Gal-3 expression is up-regulated in testicular germ cell tumours [129]
Galectin-3 is considered a marker of aggressiveness in TC [130]. While the gene is down-regulated in normal adult testis, higher amount was observed for seminomas GCT and much higher for non-seminomas GCT tissue. A higher galectin-3 mRNA was observed in non-seminomas GCT cell line (NCCIT) compared to seminoma GCT cell line (JKT1) [130].”
Finally, 2 new figures were added into the revised version of the manuscript. Fig. 4 was redrawn using info provided in ref. 119 and Fig. 5 was taken from ref. 123. Please see the file “Revised manuscript with changes highlighted.docx” showing all changes made in the revised version of the manuscript.

Round 2
Reviewer 1 Report
In the revised manuscript, the authors explained the currently description and classification of testicular cancer (TC) biomarkers and pointed out that two of the most important ones are not produced solely due to TC. Furthermore, they showed protein glycoprofiling as a promising biomarker for TC diagnostics by extracting and referring some relevant glycan analysis.
This manuscript is a resubmission of an earlier submission. The following is a list of the peer review reports and author responses from that submission.
Round 1
Reviewer 1 Report
This review article provides a comprehensive way in applying protein glycoprofiling as biomarkers of applying for disease diagnostics, monitoring and recurrence evaluation of testicular cancer. It is straightforward, well written, and concise and has clear conclusion. Definitely deserves to be published and is a valuable contribution to the International Journal of Molecular Sciences. Some minor issues could be addressed before publication.
Major points:
In abstract, it was described as three parts (the first part of the review deals with limitation of current TC biomarkers et al.), but in the text, it showed in ‘1. Testicular cancer’, ‘2. TC biomarkers’, et al. Please, keep the three parts and put these ‘2. TC biomarkers’ and so on under the three parts.
Minor points:
Put Table 1 between line 76 and line 77. The samples in line 169-171 (ref 56) were too few. It’s good to point it out at the end of this paragraph. Also in line 250-252 (ref 70 and 71), there were too few samples. But why did you use these references with very few samples? In supplementary file, ‘types of hCG: A) Binding to TGFβ-II receptor (hCG-H or hCGβ-H)’. Why use ‘or’ not ‘and’?Reviewer 2 Report
In this review, the authors well described the limitation of current testicular cancer biomarkers and expected the glycoprofiling of hCG/AFP might be novel testicular cancer biomarkers for diagnostics and prognostics purposes.
In the field of testicular cancer, the studies are few and the discovery of novel clinical biomarkers would clearly help the early detection and the monitor of the disease. The expectation in this review is provided some conclusive data for identification of biomarkers in testicular cancer. The authors cite the number of facts about the performance of glycosylated hCG/AFP as testicular cancer biomarkers, however, research data reported so far are rather limited. The studies of glycosylated hCG are investigated in low number of samples (n=1~3) and the conclusions are confusion. The researches on performance of glycosylated AFP are including some case-reports and are outdated. This review may be helpful for the design of further research in this particular area, but the available material is too limited to reach any solid conclusions. It cannot be recommended the article for publication in the International Journal of Molecular Sciences.